

# Niger discharge from radar altimetry: Bridging gaps between gauge and altimetry time series

Stefan Schröder[1], Anne Springer[1], Jürgen Kusche[1], Bernd Uebbing[1], Luciana Fenoglio-Marc[1],
Bernd Diekkrüger[2], and Thomas Poméon[2,3]

[1]Institute of Geodesy and Geoinformation, University of Bonn
[2]Department of Geography, University of Bonn
[3]Now at Agrosphere Institute (IBG-3), Forschungszentrum Jülich, 52425 Jülich, Germany

**Correspondence:** Stefan Schröder (s7stschr@uni-bonn.de)

**Abstract.** The Niger river represents a challenging target for deriving discharge from spaceborne radar altimeter measurements, in particular since most terrestrial gauges ceased to provide data during the 2000s. Here, we propose to derive altimetric rating curves by 'bridging' gaps between time series from gauge and altimeter measurements using hydrological model simulations. We show that classical pulse-limited altimetry (Jason-1 and-2, Envisat, Saral/Altika) subsequently reproduces discharge well and enables continuing the gauge time series, albeit at lower temporal resolution. Also, SAR altimetry picks up quite well the signal measured by earlier altimeters and allows to build extended time-series of higher quality. However, radar retracking is necessary for pulse-limited altimetry and needs to be further investigated for SAR. Moreover, forcing data for calibrating and running the hydrological models must be chosen carefully. Furthermore, stage-discharge relations must be fitted empirically and may need to allow for breakpoints.

*Copyright statement.*

## 1 Introduction

The Niger river, shared among Nigeria, Mali, Niger, Benin, and Guinea, represents the 14th largest river in the world, with a length of 4180 km. The Niger basin covers an area of 2.1 mio km$^2$ and provides water resources to more than 100 million inhabitants (Oyerinde et al., 2017). Mean annual discharge into the Niger Delta and the tropical Atlantic Ocean amounts to

15 5600 m$^3$ s$^{-1}$, with peaks during September reaching 27600 m$^3$ s$^{-1}$ and low-flow during winter/spring down to 500 m$^3$ s$^{-1}$ (Abrate et al., 2013). Seasonal variations are largely driven by the monsoon during June-August. During the wet season, the vast wetlands of the Inner Niger Delta with 36.000 km$^2$ regularly turn into a large lake, forming a unique ecosystem. However, inter-annual variability is large and decreased rainfall predominantly during the 1960s to the early 1980s had led to droughts and famines, while floods have occurred more frequently during the last 25 years, leading to loss of life, infrastructure damage,

and tremendous economic costs.



It is thus of obvious importance to water managers, planners and scientists to better understand and quantify Niger flows, both at short timescales with near-real time latency, and at longer timescales where discharge responds to climate and land use change (Coulthard and Macklin, 2001; Legesse et al., 2003). At the largest spatial scale, discharge measurements would be required to close terrestrial water budgets with observed or reanalysis precipitation and evapotranspiration data sets and

total water storage variations observed with the GRACE satellite mission (Springer et al., 2017), and to improve estimates of freshwater forcing for understanding ocean dynamics (e.g., Papa et al., 2012). However, the gauge observation network along the Niger is not well developed in many locations, due to periodical damage during floods, poor funding for maintenance, and armed conflict or unrest in some regions, or data is not automatically transmitted. As in most of Africa, the majority of stations ceased to provide daily discharge time series to global databases in the early 2000s.

Spaceborne radar altimetry, originally designed to monitor the world's oceans, has been suggested for long as a means to complement the declining gauge network (Koblinsky et al., 1993). The altimetry community has developed techniques to extract water levels from reprocessed ('retracked') radar echoes with uncertainties down to few cm for large lakes and few dm to about 1 m for rivers depending on width (Biancamaria et al., 2017). Radar altimetry is hampered by the long repeat cycles of the satellites (generally 10 days and longer), the coarse resolution due to groundtrack spacing, and the large footprints of the

altimeters. However, recent missions such as CryoSat-2 and Sentinel-3 have been shown to be able to capture much smaller rivers due to their improved SAR (Synthetic Aperture Radar) Delay-Doppler measuring systems. Both accuracy and precision are improved compared to classical altimetry (Fenoglio-Marc et al., 2015; Dinardo et al., 2017). For crossings of large rivers, operational altimetric level time series are provided as 'virtual tide gauges' via public data bases such as Hydroweb (Crétaux et al., 2011) or DAHITI (Schwatke et al., 2015).

Yet, radar altimeters measure water levels, and for converting them to discharge it is generally required to have a daily discharge time series from a real gauge near the virtual gauge – possible distances strongly depend on the river morphology – for an overlapping period of time. In the Niger basin, the largest obstacle to exploiting radar altimetry is that very few gauge time series are available nowadays. In fact, the only altimeter that provides a temporal overlap with the gauge time series is Topex/Poseidon launched in 1993. However, Topex/Poseidon measured with a groundtrack spacing of 270-300 km in West

Africa, and water levels have lower accuracy compared to contemporary satellites due to less accurate on-board tracking as well as ionosphere and troposphere corrections (Uebbing et al., 2015). Moreover, due to changes in river morphology we can expect that stage-discharge relations based on data from the 90s may not well be applicable to contemporary data.

In recent years, several approaches have been developed to convert radar-altimetric water levels into discharge, see Tarpanelli et al. (2013) or Paris et al. (2016) for an extended discussion. However, most of these techniques assume that a stage-discharge

('rating-curve') relation can be derived empirically and they can thus not be applied to the Niger river directly. Tarpanelli et al. (2017) have, for the Niger-Benue river, suggested to forecast flood discharge from altimetric water levels, MODIS river width, and rating curve calibration; however with in situ measurements of water levels available. Others have proposed to assimilate altimetric levels into elaborate hydrodynamic modelling (Munier et al., 2015); however such models are not always available. Therefore we propose to combine simplified hydrological models with radar altimetry. The calibrated models serve to 'bridge'



time series between gauge and altimeter era, and stage-discharge relationships are then derived using simulated discharge and altimeter data from four different missions. Our results show that altimetry subsequently can reproduce (simulated) discharge very well, and effectively continue the gauge time series, albeit at lower temporal resolution. However, we will show that (1) a careful choice of model forcing data sets is important, (2) radar retracking is key for obtaining meaningful time series (we

5 have created virtual stations which either cannot be obtained from public databases or became available only very recently), and (3) fitted empirical stage-discharge relation may need to allow for breakpoints, where the river regime changes e.g. due to riverbank overflow.

This paper is organized as follows: In sect. 2 we present the gauge, altimetry, and precipitation data that we use, and our methods for discharge conversion. Section 3 contains results and statistics, while sect. 4 concludes with a discussion and an

10 outlook.



## 2 Methods and Data

### 2.1 Study area and gauge data

We focus on the Upper Niger (Sahelian) region shown in Fig. 1, which extends from Koulikoro (Mali) to Kandadji (Niger) and includes the Inner Niger Delta. Rainfall is typically around 800 mm $a^{-1}$. Hydrographs at Koulikouro exhibit sharp peaks

around mid-September, and are affected by operating the Selingue dam on the Sankarani River, a tributary of the Niger in Southern Mali. Water moving along the Niger floods up to 25,000 $km^2$ of the inner delta during wet years and 2000 $km^2$ during dry years (Ibrahim et al., 2017). Downstream the inner delta, hydrographs are significantly flattened (e.g., Olomoda, 2012) and peak discharge is delayed (e.g., Aich et al., 2014).

We select five gauging stations for this study (Koulikoro, Dire, Koryoume, Ansongo and Kandadji), based on the following

criteria: (1) availability of daily discharge measurements, (2) temporal overlap with the data required to force our simple hydrological model, (3) distance to an altimeter crossing, and (4) minimum width of the river and crossing angle with respect to the altimeter track. Among the five stations, Koulikoro is the only one upstream the inner delta and has the highest discharge. Dire is located in the Inner Niger Delta and Koryoume right downstream of it. From Koryoume, the Niger flows 700 km until it reaches Ansongo and then approaches the country Niger, where the Kandadji station is located. The sub-basins upstream to

these gauges, for which we calibrate and run the simple lumped hydrological models (see section 2.4), are shown in Fig. 1 with purple lines.

Figure 1 includes altimeter groundtracks and the locations of virtual gauges that we created (see section 2.2) for the Envisat, Jason-1 and -2, and Saral/Altika satellite altimeters close to the five mentioned gauges. Water level data from Envisat and Saral/Altika became available very recently in the DAHITI database (Schwatke et al., 2015) close to all stations except Dire. It

is used here only for validation. We have also generated recent water level time series from Sentinel-3A (launched 2/2016) data. A Sentinel-3A virtual gauge is located about 40 km upstream of Koulikoro; this crossing almost coincides with the Envisat pass 646 crossing. The second Sentinel-3A virtual gauge that we generated is close to Koryoume, about 20 km upstream the Envisat pass 459 crossing.

Daily gauge time series are available via public archives since 1975 (Kandadji) and earlier and extend up to 2001 for Ansongo

and Koryoume, 2002 for Kandadij, 2003 for Dire, and 2006 for Koulikoro, albeit with gaps. Figure 2 shows data availability and overlap periods for the gauges, altimeters, and the model simulation. We used discharge data from the Global Runoff Data Centre (GRDC, 56068 Koblenz, Germany), and begin our analysis in 1988, since no reliable model forcing data is available prior to this date (see section 2.4). It is unknown, however, on which stage-discharge relations these discharge data are based.

### 2.2 Deriving altimetric water levels

Radar altimeters map water levels by continuously emitting microwave pulses, whose nadir echoes are recorded and digitized on-board the satellite. From these 'waveforms' one derives signal travel-time and range as measured from the antenna to the





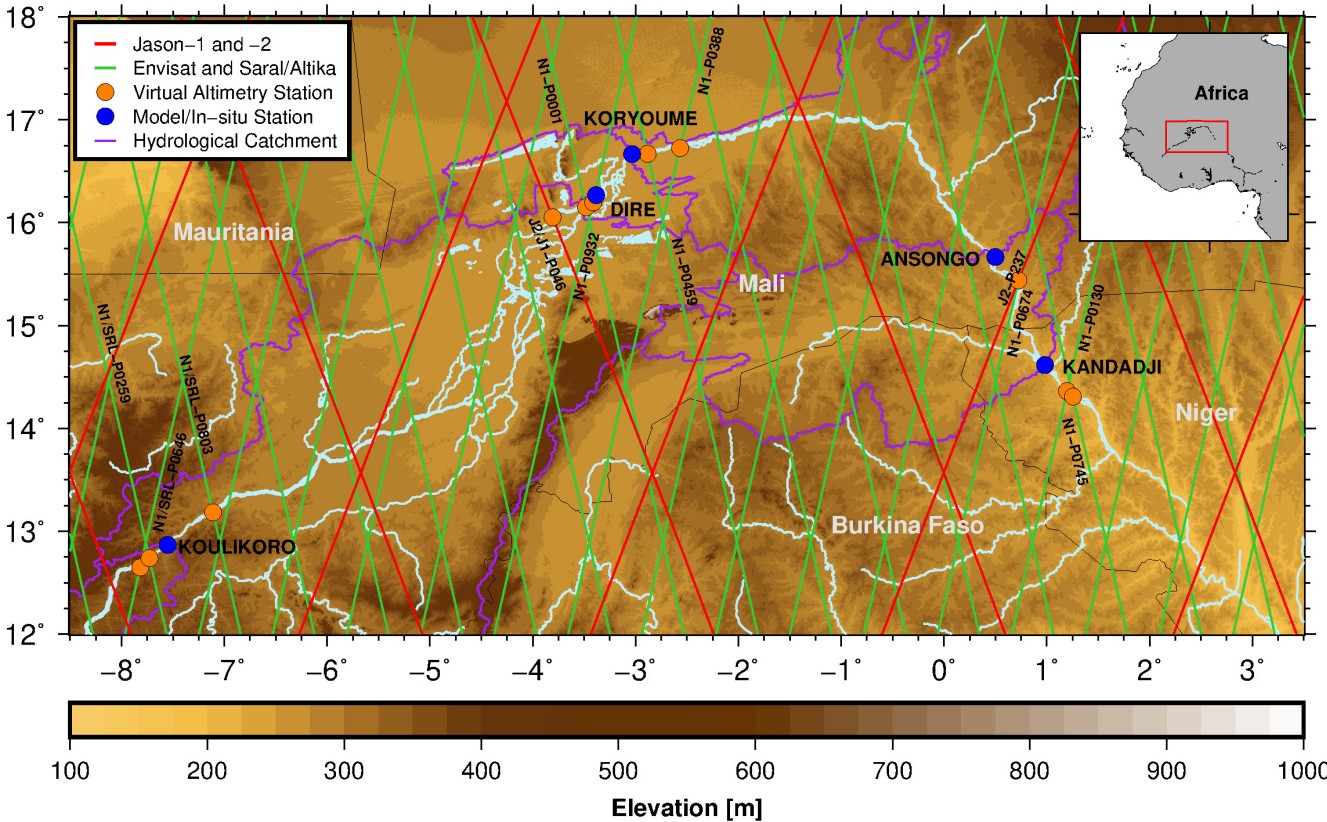

**Figure 1.** Study area, gauge and virtual gauge locations, altimeter groundtracks, and sub-basin delineations (Area: Koulikoro 118400 km$^2$, Dire 362000 km$^2$, Koryoume 378800 km$^2$, Ansongo 530000 km$^2$, Kandadji 596400 km$^2$)

water surface. Dense water level profiles across river sections from one overflight at time $t$ are then usually averaged into a single 'gauge level' $H(t)$. The Jason-1 (2001-2013) and -2 (2008-) satellites have mapped water bodies with a 10-day repeat period and inter-track spacing of about 290 km in our study area. Jason-1 and -2 followed Topex/Poseidon (1992-2006), but carried improved altimeter payloads. In the mean time, Jason-3 (launch 2016) continues this data set and Jason-CS/Sentinel-6

5 (anticipated launch 2020) will take over in time. Relative altimeter errors (i.e. with respect to an arbitrary vertical reference) are thought to be at the level of 70-80 cm Root Mean Square Error (RMSE) e.g. for Jason-2 (Tourian et al., 2016). In addition, we used Envisat (2002-2013) and Saral/Altika (2013-) to benefit from their much higher spatial resolution (about 70 km in the area), but these satellites have repeat cycles of 35 days. Relative errors are believed to be at the 60-70 cm RMSE level (Tourian et al., 2016). Absolute errors of altimetric water levels are generally larger, due to biases in altimeter calibration, retracker

10 biases and reference system effects.

In this study, we used the Jason-1/-2, Envisat and Saral/Altika 20 Hz data from the Sensor and Geophysical Data Record (SGDR) products, provided by Aviso (ftp://avisoftp.cnes.fr/AVISO/pub/) and ESA (https://earth.esa.int/) with latency of around


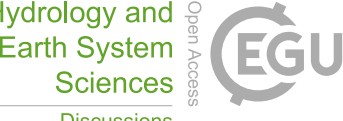


**Figure 2.** Time periods of available data; Blue: Simulated discharge; Black: Observed discharge from the GRDC; The other colours represent data periods for four different altimetry missions that are used in this work. Sentinel-3 altimetry data are available since early 2016 and used for comparison only here.

30 days. We applied corrections for microwave signal delay due to the dry troposphere (ERA-Interim), wet troposphere (ERA-Interim) and ionosphere (Nic09), and for time-varying water level changes due to solid earth tides, pole tides, and (ocean) loading tides (GOT4.10).

We 'retrack' individual radar echoes received along the river crossings of the satellites following Roscher et al. (2017), which

5  leads to much more useful ranges as compared to ranges obtained from the on-board tracker or from standard retrackers. The signal returns from the Niger river are significantly stronger compared to the returns from the surrounding land surface, and consequently the altimeter will 'see' the river off-nadir when the satellite approaches or departs from the actual cross-over location. This leads to the so-called 'hooking' effect (da Silva et al., 2010; Boergens et al., 2016), which masks water levels





with a spurious parabolic profile in the along-track surface height measurements. To remove the hooking effect, we explore the water level 'point cloud' (e.g. Fig. 3, A), which is derived as a byproduct of the STAR retracking method described in Roscher et al. (2017). The point cloud represents several possible surface heights for each measurement location; this is in contrast to other retracking techniques where typically a single best height estimate is provided. Then, for each cross-over profile, we

remove a 'hooking parabola' (Fig. 3, A) by fitting a second-order polynomial to the point clouds from our retracker by using the Random Sample Consensus (RANSAC) method (Fischler and Bolles, 1981). Due to the large number of 'likely' water levels contained in the point clouds, it is possible to detect multiple hooking parabolas (Fig. 3, A) and to remove the hooking effect even over narrow river crossings, smaller than 100m. The final water level is then derived from the peak of the parabola. For wide river crossings, where several height measurements are located over the river itself, we derive the final height from

simple averaging.

Sentinel-3 data are available since about March 2016 and we use these data here for comparison to Koulikoro and Koryoume water levels derived from earlier altimeters. We use level 2 SAR data made available via the Copernicus Open Access Hub (https://scihub.copernicus.eu) and through ESA's G-POD SARvatore Service (https://gpod.eo.esa.int/services/cryosat_sar). In these analyses the Offset Centre of Gravity (OCOG) retracker was applied to the SAR waveforms in the first dataset, whereas

the SAMOSA+ retracker (Dinardo et al., 2017) was used for GPOD data. A Hamming windows was applied, which allows noise reduction (Moore et al., 2018). The hooking effect is thought to be negligible in SAR due to the smaller footprint, and since only across-track off-ranging will contribute to this error. Moreover, SAR echoes are more accurate compared to conventional altimetry due to the multi-looking property. Whether waveforms originate from water or land reflections is decided based on a static map; this should be improved in the future. At both Sentinel-3 crossings, the river width is about 400-500 m

and the altimeter pass is about 700 m wide.

## 2.3    Stage-discharge relations

Stage-discharge relations represent the hydraulic behaviour of a river channel section, thus change with changing river morphology, and must generally be considered as unknown. Since the river banks are not vertical and the water flows faster at high stages, the relation is not be linear. The most frequently used empirical expression for the stage-discharge relation is the simple

rating curve (Lambie, 1978)

$$Q = a \cdot h^{b}. \tag{1}$$

In the above, $Q(t)$ represents the discharge in m$^3$ s$^{-1}$ and $h(t)$ is the river depth in m. In Eq. (1), $a$ (in m$^{4/3}$ s$^{-1}$) and the dimensionless parameter $b$ describe the hydraulic behaviour. These parameters can be computed from Manning's equation under idealized conditions (Paris et al., 2016). As a rule, a wide river leads to a large $a$, and shallow river banks lead to a

large $b$. However, river width is difficult to observe from space, and other characteristics like river cross-section and slope remain unknown, so the operational solution is that $a$ and $b$ are fitted to discharge and stage data observed during a calibration campaign.





**Figure 3.** Hooking effect. (A) STAR pointcloud from retracking all available Envisat cycles of P0259 crossing the Niger river. The main hooking parabola corresponding to the main river is marked in orange. (B) Virtual station of Envisat P0259 crossing the Niger river. The Shortwave Infrared (SWIR) information has been extracted from Band 6 of a Landsat-8 image captured at 2018-10-31.

Assuming gauge and virtual gauge data from altimetry are available during an overlap period, it is possible to estimate the rating curve parameters $a$ and $b$. However, spaceborne altimeters observe heights with respect to a global reference frame, which is realized through satellite orbit determination, while Eq. (1) requires water depth $h$ as measured with respect to the riverbed. Therefore, Eq. (1) is reformulated as in Chin et al. (2001) and Kouraev et al. (2004):

$$Q = a \cdot (H - Z_0)^b. \tag{2}$$

The water depth is partitioned into the water level or elevation $H$ observed with the altimeter, and the elevation $Z_0$ of the river bed, i.e. the elevation of zero flow. $Z_0$ needs to be calibrated alongside with $a$ and $b$.

The three parameters are obtained by applying a Monte Carlo approach. For any given $Z_0$, parameters $a$ and $b$ are estimated




from observed pairs of $Q$ and $H$ via minimizing the sum of squared residuals of a the linear regression model, which reads after log-transformation (Chin et al., 2001; Leon et al., 2006),

$$\ln(Q) = \ln(a) + b \cdot \ln(H - Z_0). \tag{3}$$

This regression is repeated for a wide range of possible $Z_0$ values, and the final set of parameters is found as the RMSE
minimizer with respect to observed $Q$.

For some gauges along the Niger, we find that a single rating curve may not sufficiently represent the observed stage-discharge relation. This is most likely due to changes in the geometry of the river bed at certain water stages. For stages above this level, the 'break point', we estimate an additional rating curve. For the Niger this is often required when the river bursts its banks. In our estimation of rating curves, possible break points are identified manually. When a break point is found, first the rating curve
for lower heights is estimated, subsequently the rating curve for higher stages (only $a$ and $b$) is estimated with the constraint to yield the same discharge exactly at the break point. Afterwards, stage and discharge are added back. The corresponding equation reads

$$Q = \begin{cases} a_1 \cdot (H - Z_0)^{b_1} \\ a_1 \cdot (H_b - Z_0)^{b_1} + a_2 \cdot (H - H_b)^{b_2} \end{cases} \quad \text{for} \quad \begin{cases} H < H_b \\ H > H_b \end{cases}, \tag{4}$$

where $H_b$ is the stage of the break point.

## 2.4 Simulating discharge

Simulating discharge in the Niger catchment using hydrological models is challenging since precipitation data sets rely on few rain gauges, and since it is difficult to determine evapotranspiration in the vast floodplains. In addition, dam operations affect discharge information about the management of the reservoirs are often not available. In order to bridge the gap between gauge and altimeter time series, two simple lumped hydrological models have been calibrated individually for each gauge. We
decided to use GR4J (Perrin et al., 2003) and HBVlight (Seibert and Vis, 2012) for this purpose, which allows to investigate the sensitivity of the approach with respect to the model choice. Furthermore, it is known that GR4J has limitations concerning the travel time within the catchment, and we will confirm that this limits its application to the Inner Niger Delta.

GR4J represents a daily four-parameter rainfall-runoff model, which has performed well in previous investigations for African river catchments (e.g., Bodian et al., 2018 and Kodja et al., 2018). Running GR4J requires area-averaged precipitation ($P$)
and potential evapotranspiration ($E$) data for the sub-basin upstream of the gauge. The model parameters $x_1$ to $x_4$ represent the maximum capacity of the 'production store', which is replenished from precipitation, the time lag between a rainfall event and its resulting discharge peak, the capacity of the routing store, and finally the catchment water exchange coefficient. The resulting discharge $Q$ at time $t$ can be written as

$$Q(t) = \int_{t-x_4}^{t} f(P, E, x_i) d\tau. \tag{5}$$



For each gauge, the $x_i$ are calibrated against the discharge time series while optimizing the RMSE. We use the first ten years of data for calibration, the remainder of the available discharge data (3 to 8 years) are then used as validation period. For both time periods, visual inspection is performed and the Nash-Sutcliffe coefficient (NSC) (Nash and Sutcliffe, 1970) is derived.

Precipitation data products differ considerably in the Niger region (Awange et al., 2015; Poméon et al., 2017). For simulating discharge with GR4J, we evaluated four different gridded, daily precipitation data products, i.e. PERSIANN-CDR (Precipitation Estimation from Remotely Sensed Information using Artificial Neural Networks-Climate Data Record, Ashouri et al., 2015), CMORPH v1.0 CRT (Climate Prediction Center Morphing Technique, Xie et al., 2011), TMPA 3B42 v7 (Tropical Rainfall Measuring Mission (TRMM) Multi-satellite Precipitation Analysis, Huffman et al., 2007) and CPC Unified Gauge-
Based Analysis of Global Daily Precipitation (Chen et al., 2008). CMORPH and TMPA are predominantly based on satellite data, bias corrected with GPCC and CPC gauge data, and available only since 1998, so they serve for comparison purposes here. PERSIANN-CDR contains 0.25° data from 1983 onwards, while CPC is available since 1979 on a 0.5° grid.

First, mean daily precipitation for the five upstream basins associated with the Niger gauges is constructed from the gridded precipitation estimates. Time series (after annual smoothing) are shown in Fig. 4. The largest differences between the individual precipitation data sets can be observed at Koulikoro, the most upstream station and thus related to the smallest catchment
area. When moving downstream (from top to bottom in the figure), the bias between the data sets becomes smaller. As the catchments associated with the downstream stations include the smaller Koulikoro sub-basin, we observed how precipitation biases tend to average out. However, most striking is a prolonged (2001-2007) period of low precipitation in the CPC time series, which becomes most obvious at Koulikoro, but can be observed for all five stations. We found that GR4J simulates
unrealistically low discharge for this time period, even at the more downstream stations. Therefore, we finally decided to use PERSIANN-CDR for calibrating GR4J. Although the time series starts in 1983, we discarded the first five years where annual means are up to 32 % lower than in the following years, in order to prevent calibrating in the drier period that lasted from the 1960s to the earlier 1980s.

For potential evapotranspirartion, we chose the CRU (Climatic Research Unit, University of East Anglia) TS v. 4.01 data set
(Harris and Jones, 2013), which contains monthly data from 1901 to 2016 on a 0.5° grid. It is based on the analysis of over 4000 individual weather station records and mostly homogenized.

As the second model, HBVlight (Seibert and Vis, 2012) was applied to simulate discharge and evapotranspiration. HBVlight represents a user friendly version of the HBV model (Bergström). HBVlight includes an automatic parameter estimation routine that uses numerous quality measures, and a Monte Carlo routine to perform automatic simulations for sensitivity analysis.
Like GR4J, HBV belongs to the class of rainfall-runoff models and consists of three main components, a snow routine (not used in this study), a soil moisture routine used for computing actual evapotranspiration and groundwater recharge, and groundwater as well as river routines to simulate discharge at the observed gauging station. HBVlight is a semi-distributed model, meaning that different elevation and vegetation zones can be considered, which is important for our study region (Poméon et al., 2017).





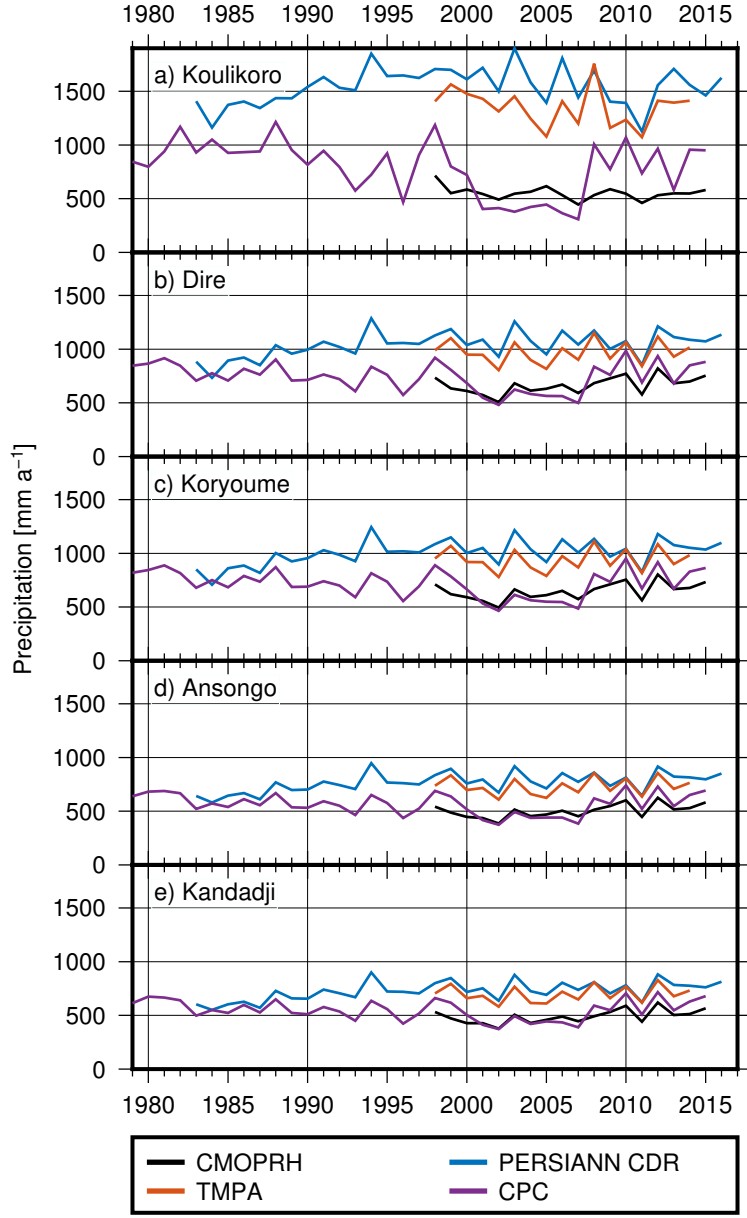

**Figure 4.** Comparison of precipitation from the datasets PERSIANN-CDR, CMORPH, TMPA and CPC for the five study catchments.

Furthermore, it offers the possibility to model lakes and can easily be adapted to the given geological situation by introducing up to three different groundwater zones. The actual version of the model is available at the website of the University of Zurich (https://www.geo.uzh.ch/en/units/h2k/Services/HBV-Model.html). It offers a higher flexibility compared to GR4J, but contains more calibration parameters (Seibert and Vis, 2012). HBVlight was applied here as a lumped model in the standard version,
5  with nine calibration parameters.



## 3 Results and Discussion

### 3.1 Simulated discharge

In Fig. 5, discharge simulated for the five Niger stations is shown together with observed discharge. For Koulikoro, Dire and Koryoume (Fig. 5a-c), observed and simulated discharge from both models are very close for most of the time (i.e. during calibration and validation periods). Even the peak flows are reproduced very well by the models. For Ansongo and Kandadji, (Fig. 5d-e), with GR4J simulated discharge appears distinctly different from the observed data, especially regarding seasonal variability.

For a more quantitative analysis, the Nash-Sutcliffe coefficient (Table 1) is computed, separately for the calibration and the validation period. As expected, the NSC is higher in the calibration period in every case except the GR4J simulation for Koulikoro, where it is almost equal. For GR4J, NSC values computed for Koulikoro, Dire and Koryoume are larger than 0.5 comfirming the good prediction skills discussed above. For Ansongo and Kandadji, the NSC of the validation period is about 0, which indicates that GR4J is not suitable here. NSC values of the HBVlight simulation are larger than those for GR4J except for the validation period at Koryoume and Koryoume.

**Table 1.** Nash-Sutcliffe coefficients for calibration and validation periods and for both models (NSC = 1 means perfect agreement between observed and simulated discharge; NSC = 0 indicates that model predictions are as accurate as the mean of the observed data; NSC < 0 indicates that the observed mean is a better predictor than the model)

|  | NSC GR4J | | NSC HBVlight | |
|---|---|---|---|---|
|  | calibration | validation | calibration | validation |
| Koulikoro | 0.57 | 0.61 | 0.87 | 0.77 |
| Dire | 0.79 | 0.74 | 0.73 | 0.65 |
| Koryoume | 0.75 | 0.50 | 0.78 | 0.35 |
| Ansongo | 0.53 | 0.07 | 0.75 | 0.69 |
| Kandadji | 0.40 | −0.03 | 0.69 | 0.59 |

### 3.2 Altimetric water level time series

Time series of river levels, which we created from retracked altimetry, are provided in Fig. 6 for the virtual stations (VS) near Koulikoro, Dire, Koryoume, Ansongo, and Kandadji. Multiple VS belong to one gauging station due to multiple ground-track/river crossings nearby. Individual time series from Envisat and Jason agree well during their overlap time periods (Dire, Ansongo). Gaps occur when no observations are available, which can happen due to 'loss of lock' of the altimeter instrument. Due to undulating terrain, the onboard tracker is then unable to follow the range and backscatter variations of the reflected echoes. Consequently, it looses track of the leading edge of the radar return, which serves as a reference for the data window that is transmitted to Earth.



**Figure 5.** Observed and simulated discharge for the five Niger stations. The first ten years of data serve as calibration period, marked by the vertical black line. The validation period starts after these ten years.





**Figure 6.** Time series of relative river heights for each investigated station. Legend composition: Satellite (N1 for Envisat, SRL for Saral/Altika, J for Jason) - pass number (P).





We find good agreement between our reprocessed time series and the Envisat mission time series from the DAHITI archive (Schwatke et al., 2015) with correlations up to 0.99 and RMS differences between 0.2 m and 0.5 m for the stations Koulikoro, Koryoume, Ansongo, and Kandadji. For Dire no external data from altimetric data bases is available for validation.

For Koulikoro, water level series from two neighboring Envisat and Saral/Altika river crossovers with a distance of about 10 km (passes 259 and 646, see Fig. 1) match quite well. At the third crossover (pass 803) about 70 km downstream (and about 40 km downstream the terrestrial gauge) the amplitude is larger by about 0.5-1 m.

Dire is located in the Inner Niger Delta, prone to frequent flooding events. It is thus a difficult area to derive river heights due to the various tributaries of the Niger river, which strongly influence the radar returns, resulting in overlapping hooking parabolas. One Jason-1/2 and two Envisat crossovers are located within a 35 km stretch, and we observe water levels with annual variability of up to 5.5 m with a RMS difference of 1.25 m between different missions and river crossovers.

For Koryoume, two Envisat river crossovers with about 35 km distance are evaluated and water levels with a RMS difference of 0.6 m between the two crossovers are observed.

Annual water level variability at Ansongo and Kandadji is with about 2 m amplitude lower compared to the more upstream stations (amplitudes of about 3 m). Albeit of differing temporal resolution, the Envisat and Jason-2 data match quite well for Ansongo since both cross the river at almost the exact same location (RMS difference of 0.25 m). For Kandadji, two Envisat crossovers at 8 km distance and with a temporal shift of 13 days provide similar water levels.

In Fig. 7, Sentinel-3A (S3A) river levels from the years 2016 to 2018 are compared to the Envisat data measured ten years earlier (2006 to 2008). For Koulikoro, the S3A measurements (different solutions shown in black and red) show a slightly longer low water period and a higher amplitude. This may well be due to river regime changes, but it could result from annual variations as well. Also, altimeter sampling effects cannot be excluded without further investigations. At the VS near Koryoume, the S3A GPOD solution (black) shows a hydrograph which is very close to the time shifted Envisat measurements. The Copernicus Hub Land solution (red) appears somewhat different with higher amplitudes and longer high water periods.

## 3.3 Altimetric rating curves and discharge

Figure 8 displays rating curves computed from simulated discharge and altimetric water levels as described in section 2.3. Figure 9 shows simulated and altimetry-derived discharge. Altimetry rating curves are derived from the full overlap period between simulated discharge and the data period of each altimetry mission, which is limited from 2002 to 2010 in case of Envisat, and limited from 2013 to 2016 in case of Saral/Altika.

For Koulikoro, altimetric discharge is derived from the Envisat and Saral/Altika missions at 35 days temporal resolution (Fig. 8a). We observed that for GR4J, the rating curve parameters differ depending on the the satellite data we use, i.e. Envisat (2002-2010, blue curve) or Saral (2013-2016, green curve). Rating curves estimated from the HBVlight simulation differ from the rating curves from GR4J, but differences between the two HBVlight rating curves are small (orange and red curve). Obvi-




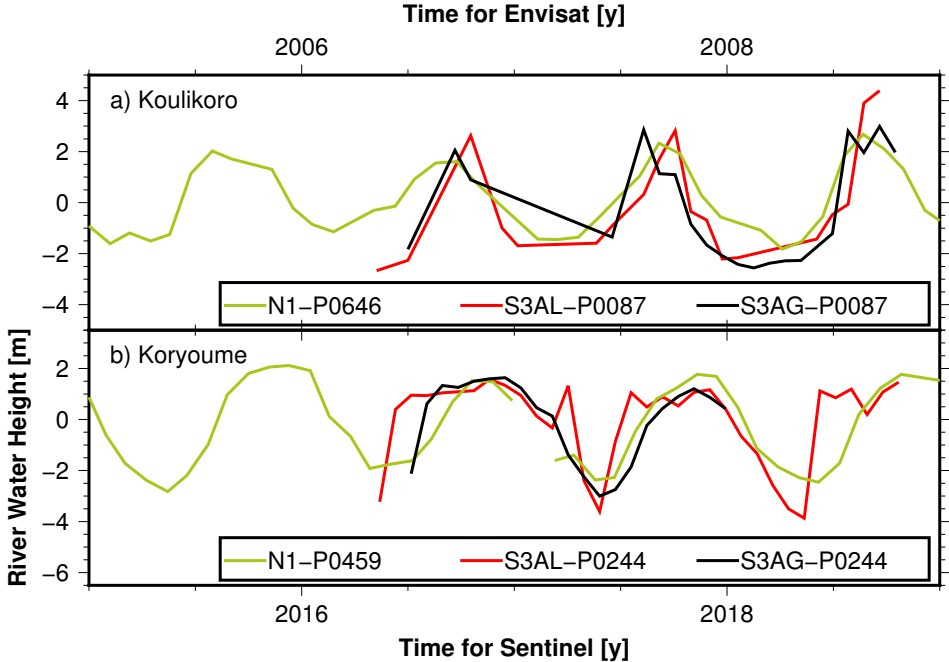

**Figure 7.** River Water height anomaly from Sentinel-3A (S3A) at two stations compared to Envisat data measured ten years before (green). The two S3A solutions are the Copernicus Hub Land (red) and the GPOD solutions (black).

ously, the choice of the hydrological model has significant impact on the estimated rating curve. Figure 9a shows that altimetric discharge peaks (dotted lines) from Envisat (2002-2010) are often lower as compared to simulated discharge (solid lines); this is expected since the stage-discharge relation is derived as a fit where we neither downweighted peak nor low flows. Also, altimetric discharge inherits the 35 day temporal resolution given by altimeter revisit cycles, and may thus simply miss peaks.

Furthermore, it is obvious that the yearly peaks of the altimetric discharge time series are less variable than the peaks from discharge simulated by the hydrological models. This was expected due to the rather uniform annual amplitudes of the water level time series (Fig. 6) and suggests that the hydrological models may overestimate such variability. For Saral/Altika, it appears the short overlapping period considered for estimating the rating curve does not lead to worse results compared to Envisat, and peaks of altimetric discharge are even closer to simulated discharge.

For Koulikoro, we have an overlapping period between observed discharge and Envisat water level time series for a period of four years. As a check of our methodology, we have estimated an alternative rating curve based on these observed data only (Fig. 8a, black curve). Again, the shortness of the period does not affect the result, measurements scatter less around the (mean) rating curve and discharge from altimetry is close to observed discharge (cf. dotted and solid black lines in Fig. 8). Low flows from altimetry appear quite realistic. Noticeable is that the rating curve is almost parallel to the two rating curves estimated

with HBVlight simulation, and only the river depth ($Z_0$) estimate is found different, which leads to the shift. The correctness of $Z_0$ is difficult to assess, but it is not of primary concern since a $Z_0$ shift in the rating curve does not affect the resulting



altimetric discharge. Thus, these results confirm the applicability of the approach of estimating rating curves from simulated discharge, at least with the HBVlight model, where the simulation works well.

At Dire (Fig. 8b), we observe for the GR4J simulation that estimating a rating curve with one break point (purple curve) indeed improves the estimation of Envisat-based discharge (the RMS difference can be reduced by 17 % when compared to a simple relation). Altimetry still misses peak simulated discharge, but the discharge hydrographs are much closer and low to medium flows fit better. For the HBVlight simulation, different parameters are estimated for the rating curves and introducing break points does not improve results. In summary, altimetry misses simulated peak flows by about 30 % but appears to reproduce the overall shape of the hydrograph well. However, comparisons against observed discharge are not possible and we do not know the truth in this case.

We observe that for Koryoume the situation is comparable to Dire; fitting a rating curve with GR4J simulation requires introducing a breakpoint and again altimetric (Envisat) discharge appears much more regular as compared to simulated one. With the HBVlight simulation we find that adding a break point does not improve results. The rating curve without break point fits well and mostly agrees to the GR4J rating curve with break point.

For Ansongo, we do not use the GR4J simulation (cf. Fig. 5). Discharge simulated by HBVlight overlaps with altimetry data from Envisat and Jason-2. The two estimated rating curves differ mostly by a $Z_0$ shift, leading to almost identical altimetric discharge. This can be seen in Fig. 9d in the overlap period of the two missions (2008-2010). Simulated and altimetric discharge exhibit RMS differences for Envisat and Jason-2 of 328 m$^3$ s$^{-1}$ and 348 m$^3$ s$^{-1}$, respectively, and NSC values of 0.56 and 0.49. The Kandadji station is omitted in this discussion due to the insufficient amount of altimetry and discharge data.

In summary, we find that relatively large scatter renders the estimation of stage-discharge relations difficult. This may have been expected due to the challenging study region. Although one expects that with higher water levels altimetry provides more reliable results (since the river is wider), then the sensitivity of changes in water level with respect to discharge is higher. This characteristic can be observed well at the scattering points in Fig. 8d. Fitted stage-discharge relations will inevitably lead to 'mean' peak and low flows.

Figure 10 visualizes the seasonal cycle of discharge for the five stations as obtained from gauge data, model simulations, and from radar altimetry. The day of peak flow is listed in Table 2. We notice that modelled peak days are generally ahead of observed peaks except for Koulikoro; this points to the problem of representing travel time in the models. Low flow and peak flow times (and peak discharge) for Ansongo and Kandaji appear to nearly coincide, this is due to the short travel time between the two stations which are only about 150 km apart. Between Dire and Koryoume (about 80 km), a phase lag of a few days is identified in gauges and models but obviously misrepresented in altimetry (cf. Table 2). When computing the mean annual hydrographs with daily available observed or simulated discharge, there are multiple values for each day getting averaged. For altimetry, this is not nessecarily the case due to the lower temporal resolution. Thus, peaks identified from altimetric data may refer to invidual years rather than to mean annual values. After correcting this effect by fitting an annual signal per virtual gauge we find the peak timings much closer to those of observed discharge.







**Figure 8.** Rating Curves for a) Koulikoro, b) Dire, c) Koryoume, and d) Ansongo. The Kandadji station is omitted due to the insufficient amount of altimetry and discharge data. The points are the discharge values plotted against the altimetric water depth. The lines are the rating curves, which are fitted through the points. *The red Rating curves are created with Saral/Altika data for Koulikoro, Jason-1 for Dire, and Jason-2 for Ansongo.





**Figure 9.** Altimetric discharge (dotted lines) together with observed and simulated discharge (solid lines); RC = rating curve; BP = break point. The Kandadji station is omitted due to the insufficient amount of altimetry and discharge data.







**Figure 10.** Mean annual hydrographs. The colour indicates the station, the line style indicates the discharge source (cf. legend).

**Table 2.** Dates of maximum flow. Altimetry avering is done by fitting an annual signal trough the points. The points and complete hydrographs can be seen in Fig. 10.

| Station name | Observed | GR4J | HBVlight | Altimetry | Altimetry (averaged) |
|---|---|---|---|---|---|
| Koulikoro | Sep 22 | Sep 28 | Oct 2 | Aug 27 | Sept 13 |
| Dire | Nov 1 | Oct 18 | Oct 8 | Aug 6 | Nov 11 |
| Koryoume | Nov 7 | Oct 30 | Oct 23 | Dec 1 | Nov 26 |
| Ansongo | Dec 11 | Nov 17 | Nov 12 | Jan 15 | Dec 8 |
| Kandadji | Dec 11 | Nov 26 | Nov 11 | Jan 3 | Nov 28 |



## 4   Conclusions

Radar altimetry enables one to observe water levels for larger rivers, although temporal resolution is generally low due to satellite revisit times. We find that careful processing of altimeter data, i.e. retracking and accounting for 'hooking' effects due to the dominant river signal at off-nadir locations, allows one to generate reliable water level time series also for river

crossovers that are not contained in public data bases, which operate automated processing chains. We found that comparisons between neighboring crossovers, i.e. from ascending and descending satellite passes and between different missions, fit usually quite well although crossovers are located up to 70 km apart. This has been observed already by others, but we can confirm it here for a quite challenging region where a braided river with often multiple but narrow channels creates multiple echoes.The Sentinel-3 SAR data pick up the signal measured by earlier altimeters quite well. We find the altimetric hydrograph flattening

out from Koulikoro to Kandadji as expected, but with little interannual variability between the years. With time, flooding and morphological changes add to altimetric noise, which appears in a range of several dm up to one meter and corresponds to what other studies found.

Since observed discharge time series generally are available only until the 2000s years, we have used simple hydrological models for simulating discharge, after station-by-station calibration. We found this approach works generally well for most

gauges. The HBVlight simulates discharge well for all gauges, while the GR4J model fails to reproduce low flows for some gauges, which is likely due to model shortcomings concerning travel time but of course also related to the specific calibration parameters. A careful choice of climate forcing data has turned out to be essential. Future research may concentrate on more sophisticated models. However, all models depend on observed precipitation, for which different data sets differ greatly.

Converting observed altimetric levels into discharge requires adopting stage-discharge relation derived at gauges. For tempo-

rally non-overlapping periods of data, where gauge and altimetric overpass may be tens of kilometers apart, deriving such a relation represents a challenging and still unsolved problem. We find that simulated discharge may aid in creating empirical altimetry-discharge rating curves, albeit it is difficult to assess the validity of the approach. Different models, although based on the same precipitation data and all calibrated, generate different rating curves. For five gauges along the central Niger, including the Inner Niger Delta, we find mixed results. Altimetry discharge exhibits generally much less interannual variability

as compared to simulated discharge; this is most likely due to problems with the observed precipitation data set. Altimetric discharge also does not capture peak flows that the model predicts while low flows fit reasonably well; this appears to be related to the temporal resolution of the satellite overpasses. We have shown that rating curves may need to account for breakpoints, most likely when the river inundates its banks, but again this depends on model simulations.

We find that, averaged over the entire study period, model simulations capture the observed timing of the annual peak flow

mostly within two weeks. Deriving these peak days from altimetry necessitates interpolating the altimetric observations, fitting an annual signal enables one to reconstruct the peak timings as close to (earlier) gauge observations as the models do.

We suggest that future research could ultimately focus on combining model simulation and model parameter estimation with gauge and multi-mission altimetry observations within data-assimilating frameworks. Remote sensing of channel width (Elmi





et al., 2015), which now provides greatly improved resolution due to e.g. Sentinel data, should be explored jointly with radar altimetry. Near real time altimetry could provide discharge with 3-5 h latency and would thus enable to utilize such frameworks for e.g. flood forecasting purposes. On the other hand, deriving consistent and long discharge time series would enable one to close budgets together with GRACE water storage data, and e.g. assess biases in reanalysis or remote sensing precipitation and

evapotranspiration data products (Springer et al., 2017).

*Data availability.*  All data – the freely available external data as well as the data that was constructed in this work – can be obtained from the authors upon request.

*Author contributions.*  Stefan Schröder, Anne Springer, and Jürgen Kusche designed the experiment. Stefan Schröder (GR4J simulation, rating curves, and altimetric discharge), Bernd Uebbing (altimetry), Luciana Fenoglio-Marc (SAR-altimetry), and Bernd Diekkrüger (HBVlight

simulation) did the computations. Stefan Schröder, Bernd Uebbing, Bernd Diekkrüger, and Jürgen Kusche wrote the paper. Anne Springer and Thomas Poméon helped with the acquisition and choice of model forcing data. All authors provided critical feedback and helped to shape the research, analysis and manuscript.

*Competing interests.*  The authors declare that they have no conflict of interest.

*Acknowledgements.*  This study has been part of the COAST project (Studying changes of sea level and water storage for coastal regions in

West-Africa using satellite and terrestrial data sets) of the University of Bonn, supported by the Deutsche Forschungsgemeinschaft (German Research Foundation) under Grants No. DI443/6-1 and KU1207/20-1. We are further grateful to the Global Runoff Data Centre in 56068 Koblenz, Germany for providing discharge data.



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
