# Peer review of "Niger discharge from radar altimetry: Bridging gaps between gauge and altimetry time series"

_Hydrology and Earth System Sciences, 2019_

## Short Comment (SC1) · 1 Feb 2019

I will not comment the content and methodology of this paper.

But I would like to point out that there was already another paper published in 2017 which investigates the river discharge of the Niger Basin by using satellite altimetry and in-situ data. Therefore, I would strongly recommend to consider and discuss the following publication in your paper.

Tourian M.J., Schwatke C., Sneeuw N.: River discharge estimation at daily resolution from satellite altimetry over an entire river basin. Journal of Hydrology, Vol. 546, 230-247, 10.1016/j.jhydrol.2017.01.009, 2017

---

## Referee Comment (RC1) · Anonymous Referee #1 · 28 Feb 2019

General Comments: In this paper, the authors present several interesting contributions to the hydrology, hydrologic modeling and river altimeter communities, with the Niger River basin as platform for case study. The authors present a method of retracking for handling the "hooking" effect that works well even on complex braided systems. The authors confirm that time series from neighboring altimeter crossovers agree well, at distances as large as 70 km. The authors demonstrate effectiveness of two different discharge models in the region, and their limitations, citing the importance of choice in forcing data. Finally they present a method of generating discharge with altimetry data where they first create a rating curve based on modeled discharge for the time period in question. For the most part I find that the work meets the standards required of HESS publications and that with a few minor revisions it will ready for publication.

[Figure]

Specific Comments: 1. Page 5 Lines 5-9: In your discussion of bias here you cite : Tourian, M., Tarpanelli, A., Elmi, O., Qin, T., Brocca, L., Moramarco, T., and Sneeuw, N.: Spatiotemporal densification of river water level time series by multimission satellite altimetry, Water Resour. Res., 52, 1140–1159" 2016. The issue is that the only discussion in this paper with regard to RMSE is in comparison of densified time series (heavily processed, not direct measurements) with gages in a specific river (the Po). I think the numbers used are from table 5. There is a section on handling relative altimeter bias with it's own table (4). The authors are careful to point out that they compared with a tide gage on at the mouth of the Po to get these values and that altimeter bias is regionally specific. It's okay to cite this information here if present the correct numbers, but only if it is noted that this is not a general rule that is broadly applicable. Alternatively the authors could cite a range of absolute errors present in the literature and avoid the bias issue entirely. 2. Page 10 Line 27-page11 Line5: I assume you are using the same forcing data here that you use to run GR44J ? I think citing a publication about your model rather than going into specifics is fine here, but you haven't explicitly stated what forcing data is used for GBVlite, and importance of forcing data is mentioned throughout the paper. 3. Page 13 lines1-5: DAHITI is a great source of data, but I find performing a fit evaluation with it to be quite odd. It's essentially comparing one non-validated set of altimetry elevations with another. Making a comparison is fine, but It should be clear to the reader that the DAHITI database is also altimeter data and in this case non-validated. Technical corrections: Roscher et al. (2017) is cited, but not included in the list of references

---

## Referee Comment (RC2) · Anonymous Referee #2 · 28 Feb 2019

Reviewer comment on manuscript Niger discharge from Radar altimetry : Bridging gaps between gauge and Altimetry time series presented by Schroder at al.

This stud aims at prividing tools for predicting discharge from altimetry data in absence of gauge data. In order to achieve this goal, authors : 1- process the raw radar echoes distributed for satellite altimetry missions, 2- run rain-discharge models in various configurations 3- compute the Height - Discharge relationships, the so-called rating curves for al the model options, 4- select the best rating curves in the dataset provided by the previous step

Actually, this work is interesting and might be published but not in the present form, mostly because of significant lacks in bibliogrpahy, and consequently, lack of comparison of their results and methods with previous ones. Also, many inaccurate statements

have to be rewritten. Introduction in particular needs an in-depth rewriting :

Specific comments :

Introduction : line 13 : wrong citation. There have been tens of paper dedicated to "developping techniques" (some of them cited in the manuscript). Biancamaria is not one of them. May be acceptable as "see review in Biancamaria et al., 2017)" ... Line 14 : Radar altimetry is not "hampered" by the groundtrack spacing of the orbits. Indeed, in many basins, the number of crossings between the satellite orbit and the river drainage network is larger that the number of gauges that have ever existed in these bassins. the fact that one cannot choose the location of the measurements is hampering radar altimetry. Radar altimetry is not "hampered" by the large footprint of the altimeters, strictly speaking. For sure, punctual measurements as the ones gained by laser technology (ICESAT) are much easier to process. The size of the footprint has drawbacks, for sure, but it also has significant advantages (the hooking effect for example as it is recalled in the present study, or the averaging effect over a large surface) Line 15 : LRM altimetry has been used to produce hundreds of series over narrow reaches, up to a few tens of meters wide (see for example those in the Amazon bassin distributed by hydroweb). I may have missed the publications (and none is given to support the statement) but I did not read that either Cryosat-2 or S3A did much better, up to now. SAR just enables sampling more small reaches than LR Mode does. Line 21 : "... it is generally required ...". "generally" is such a vague wording that one cannot say it is wrong but "a real gauge near the virtual gauge" is not required. Water levels have been converted into discharges in many studies, using a variety of méthodologies, including the present one. Line 29-30 : From the sentence "However ....", I understand that the rating curves cannot be applied to the Niger river. Yet, it is the focus of the study ??? I guess that the sentence should be rewritten (maybe stating that "single-polynom" rating curves cannot be applied ... ?? Line 34 : What about the works in Penidotti et al. (HESS, 2012), Casé et al. (HESS, 2016), Tourian et al. (HESS 2017) or Fleishman et al. (JoH, 2018) ?? These models are fully available, in particular the discharge series recently computed by Fleichman et al. (2018) using MGB and satelite altimetry in exactly the same area Page 2, line 4 : that "radar retracking is key for obtaining meaningful time series..." has been shown in many publications over more than a decade ...

Methods and Data Page 5 line 5 : Capability in retrieving water levels has considerably evolved from TP, Jason1 to Jason 2. Hence, they cannot be qualified by a single "70-80 cm RMSE". Note that much better results than those by Tourian et al. (2016) can be found in the literature for Jason 2 (see for instance Seyler et al., 2012) Line 8: Tourian et al. (2016) estimated their baises over the ocean, which is of limited meaning for rivers because the echoes are too different. other estimates are available in the litterature Linbe 9-10 : "biases [...] reference systems effects". Could the authors explain which "reference system " may turn into significant biais effect ?? Page 6 lines 4-5 : Roscher et al. (2017) retracked altimetry data on the coastal domain. I may have missed the publication but I am not aware of any publication showing that their STAR retracking performs better over rivers (both in height accuracy and number of data) than the other -usual- algorithms, such as ICE1, ICE3, ... . If so, authors should add a reference. Line 8 : Berry presented the hooking effect extensively in meetings in the early 2000's but as far as I know, the first publication presenting the use of these parabolas to gain accuracy in retrieving water levels is Frappart et al. (RSE 2006). Line 8: "which masks water levels". Actually, since the onboard tracking is locked on the river echo, the bank topography is masked by the river, not the opposite. Page 7 line 8 : "... even over narrow river crossing ...". It may be a question of my own limitations in english but I think that "even" should be replaced by "in particular" sine the hooking effect is particulary visible -and useful- over the narrow reaches. Line 20 : Authors mention that the S3A data are distributed with 2 retracking algorithms, the OCOG and the Samosa ones. I suggest that both algorithms will be tested in their study. Line 26 : Aiuthors should mantion that the unit of a is m**(4/3) /s only if b=5/3 ( if b is different from 5/3, the units would not equilibrate on both sides of Eq 1 Line 30 : "river width is difficult to observe from space". This sentence should be reworded since river width is definitely

not difficult to observe from space and it is commonly derived from existing satellite imagery products Page 8, line 1 : "assuming gauge and virtual gauge data [...] are available...". Replace "gauge" by "discharge" ?? Page 9 Eq 3 : I suggest that authors mention that Guetirana & Peters-Lidar (HESS 2013) showed that this methology based on minimum of squared residuals of Log regressions can converge towards physically meaningless values (which may explain the final results in terms of rating curves found in the present study Line 9 : I suggest that the authors present an example of "break point" in a figure. Page 17 : all these results, although interesting, lack analysis with respect to literature, in particular with respect to the recent study by Tourian et al. (HESS 2017) and by Fleishman et al. (JoH, 2018). Page 18: Figure 8 is quite difficult to read and I suggest that authors redo it. Actually, it would be more clear to plot the discharge versus the true altimetric height instead of versus the water depth (given that, in addition, no information is provided about the Zo parameters). Plotted this way one would have only 2 discharge values for a given altimetric height. Also, information about the Zo parameters could be presented simply by the intersection of the rating curves with the height axes. Page 21, line 21 : Sentence "we find that simulated discharge may aid..." is really over-selling the findings of the present study since many studies already showed such a result. The sentence has to be reworded. Line 28 : ".... most likely...". In absence of any evidence presented in the study, I think that "presumably" would be better adapted.

---

## Author Comment (AC1) · 23 Apr 2019

We thank referee 1 for her/his time and effort and for providing constructive comments.

**RC:** General comment: In this paper, the authors present several interesting contributions to the hydrology, hydrologic modeling and river altimeter communities, with the Niger River basin as platform for case study. The authors present a method of retracking for handling the "hooking" effect that works well even on complex braided systems. The authors confirm that time series from neighboring altimeter crossovers agree well, at distances as large as 70 km. The authors demonstrate effectiveness of two different discharge models in the region, and their limitations, citing the importance of choice in forcing data. Finally they present a method of generating discharge with altimetry

data where they first create a rating curve based on modeled discharge for the time period in question. For the most part I find that the work meets the standards required of HESS publications and that with a few minor revisions it will ready for publication.

**AR:** Thank you for the structured overview. Your comments were very helpful. We provide our responses and respective changes below.

**RC:** Comment 1: Page 5 Lines 5-9: In your discussion of bias here you cite : Tourian, M., Tarpanelli, A., Elmi, O., Qin, T., Brocca, L., Moramarco, T., and Sneeuw, N.: spatiotemporal densification of river water level time series by multimission satellite altimetry, Water Resour. Res., 52, 1140–1159" 2016. The issue is that the only discussion in this paper with regard to RMSE is in comparison of densified time series (heavily processed, not direct measurements) with gages in a specific river (the Po). I think the numbers used are from table 5. There is a section on handling relative altimeter bias with it's own table (4). The authors are careful to point out that they compared with a tide gage on at the mouth of the Po to get these values and that altimeter bias is regionally specific. It's okay to cite this information here if present the correct numbers, but only if it is noted that this is not a general rule that is broadly applicable. Alternatively the authors could cite a range of absolute errors present in the literature and avoid the bias issue entirely.

**AR:** Thank you for pointing out the details behind the numbers we cited. The author is right in saying that the "70-80cm" RMSE is not a general rule. We decide to cite a range of errors coming from more than just this one study. In alignment with the comment of reviewer 2 we modify the sentence to "Relative altimeter errors (. . .) are thought to be at the level of 20-80 cm Root Mean Square Error (RMSE) e.g. for Jason-2 dependent on river width (Papa et al., 2012, Seyler et al., 2013, Tourian et al., 2016)".

**RC:** Comment 2: Page 10 Line 27-page11 Line5: I assume you are using the same forcing data here that you use to run GR44J ? I think citing a publication about your model rather than going into specifics is fine here, but you haven't explicitly stated

what forcing data is used for GBVlite, and importance of forcing data is mentioned throughout the paper.

**AR:** Thanks for pointing out that we missed stating which forcing data we used for HBVlight. We used the same forcing data as for GR4J. We add the sentence "We used the same forcing data and calibration period as for GR4J." after "As the second model, HVBlight..."

**RC:** Comment 3: Page 13 lines 1-5: DAHITI is a great source of data, but I find performing a fit evaluation with it to be quite odd. It's essentially comparing one non-validated set of altimetry elevations with another. Making a comparison is fine, but It should be clear to the reader that the DAHITI database is also altimeter data and in this case non-validated.

**AR:** It is true that both our time series and the DAHITI time series are unvalidated and we cannot know the truth in that case. We add the sentence: "This does not validate our time series but it gives a measure of how much impact the retracking has."

**RC:** Technical corrections: Roscher et al. (2017) is cited, but not included in the list of references

**AR:** We added the reference.

**References**

- Papa, F., Bala, S.K., Pandey, R.K., Durand, F., Gopalakrishna, V.V., Rahman, A., Rossow, W.B., 2012. Ganga-Brahmaputra river discharge from Jason-2 radar altimetry: An update to the long-term satellite-derived estimates of continental freshwater forcing flux into the Bay of Bengal. Journal of Geophysical Research: Oceans 117. https://doi.org/10.1029/2012JC008158

- Roscher, R., Uebbing, B., Kusche, J., 2017. STAR: Spatio-temporal altimeter waveform retracking using sparse representation and conditional random fields. Remote Sensing of Environment 201, 148–164. https://doi.org/10.1016/j.rse.2017.07.024

- Seyler, F., Calmant, S., Silva, J.S. da, Moreira, D.M., Mercier, F., Shum, C.K., 2013. From TOPEX/Poseidon to Jason-2/OSTM in the Amazon basin. Advances in Space Research, Satellite Altimetry Calibration and Deformation Monitoring using GNSS 51, 1542–1550. https://doi.org/10.1016/j.asr.2012.11.002

- Tourian, M., Tarpanelli, A., Elmi, O., Qin, T., Brocca, L., Moramarco, T., and Sneeuw, N.: Spatiotemporal densification of river water level time series by multimission satellite altimetry, Water Resour. Res., 52, 1140–1159, 2016. https://doi.org/10.1002/2015WR017654.

---

## Author Comment (AC2) · 23 Apr 2019

We thank referee 2 for her/his time and effort and for providing constructive comments.

**RC:** General comment: This study aims at prividing tools for predicting discharge from altimetry data in absence of gauge data. In order to achieve this goal, authors : 1- process the raw radar echoes distributed for satellite altimetry missions, 2- run rain-discharge models in various configurations 3- compute the Height - Discharge relationships, the so-called rating curves for al the model options, 4- select the best rating curves in the dataset provided by the previous step. Actually, this work is interesting and might be published but not in the present form, mostly because of significant lacks in bibliography, and consequently, lack of comparison of their results and methods with

previous ones. Also, many inaccurate statements have to be rewritten. Introduction in particular needs an in-depth rewriting.

**AR:** Thank you for the overview and highlighting the strengths of the manuscript. The bibliography and comparisons have been significantly extended, and several statements have been rewritten to be more focused. We provide detailed responses below.

**RC:** Introduction : line 13 : wrong citation. There have been tens of paper dedicated to "developping techniques" (some of them cited in the manuscript). Biancamaria is not one of them. May be acceptable as "see review in Biancamaria et al., 2017)"

**AR:** Biancamaria et al. (2017) indeed provides a review. We have modified the citation as suggested.

**RC:** Line 14: Radar altimetry is not "hampered" by the groundtrack spacing of the orbits. Indeed, in many basins, the number of crossings between the satellite orbit and the river drainage network is larger that the number of gauges that have ever existed in these basins. The fact that one cannot choose the location of the measurements is hampering radar altimetry. Radar altimetry is not "hampered" by the large footprint of the altimeters, strictly speaking. For sure, punctual measurements as the ones gained by laser technology (ICESAT) are much easier to process. The size of the footprint has drawbacks, for sure, but it also has significant advantages (the hooking effect for example as it is recalled in the present study, or the averaging effect over a large surface) .

**AR:** Whether the groundtrack spacing "hampers" analyses depends on the geographical orientation of the river (East-West is better than North-South) and, of course, on what one considers as "sufficient" spacing". We agree with the reviewer that for our study region the number of crossings is larger compared to the number of gauges, and we have thus removed this statement. With regards to the size of the footprint, there are advantages and disadvantages that come with smaller footprints such as in laser (or SAR) altimetry. Yet, for braided rivers in complex terrain, in the presence of lakes

and/or inundated areas close to the river, we believe a smaller footprint would be of advantage. We changed the sentence to "Radar altimetry is hampered by the long repeat cycles of the satellites (generally 10 days and longer) and the large footprints of the altimeters renders the processing less straightforward as compared to later altimetry".

**RC:** Line 15 :LRM altimetry has been used to produce hundreds of series over narrow reaches, up to a few tens of meters wide (see for example those in the Amazon basin distributed by hydroweb). I may have missed the publications (and none is given to support the statement) but I did not read that either Cryosat-2 or S3A did much better, up to now. SAR just enables sampling more small reaches than LR Mode does.

**AR:** The reviewer is correct, LRM altimeters have been able to measure reaches of a few tens of meters width (under favourable conditions). We changed the sentence to "However, recent missions such as CryoSat-2 and Sentinel-3 have been shown to be able to capture more small river reaches due to their improved SAR (Synthetic Aperture Radar) Delay-Doppler measuring systems". We also changed the following sentence (line 17) "For crossings of large rivers . . . via public data bases. . ." to "For crossings of medium and larger rivers. . .".

**RC:** Line 21 : "... it is generally required ...". "generally" is such a vague wording that one cannot say it is wrong but "a real gauge near the virtual gauge" is not required. Water levels have been converted into discharges in many studies, using a variety of méthodologies, including the present one.

**AR:** We changed the sencence to "and the most straightforward way for converting them to discharge requires to have a daily. . .". "Generally" referred to "in most studies" in the original text but we agree this may be debated. But it cannot be doubted that the rating curve approach is most straightforward, see. Paris et al or Tarpanelli et al that we cite in line 28.

**RC:** Line 29-30 : From the sentence "However ....", I understand that the rating curves cannot be applied to the Niger river. Yet, it is the focus of the study ??? I guess that

the sentence should be rewritten (maybe stating that "single-polynom" rating curves cannot be applied ... ??

**AR:** What we meant here is that we don't have overlap periods between gauge time-series and altimetry and the RC cannot be derived "directly". We clarify this now by modifying the sentence to "However, most of these techniques assume that a stage-discharge ('rating-curve') relation can be derived empirically during an overlap period and they can thus not be applied to the Niger river directly."

**RC:** Line 34 : What about the works in Penidotti et al.(HESS, 2012), Casé et al. (HESS, 2016), Tourian et al. (HESS 2017) or Fleishman et al. (JoH, 2018) ?? These models are fully available, in particular the discharge series recently computed by Fleichman et al. (2018) using MGB and satelite altimetry in exactly the same area

**AR:** Thanks for pointing these works out. Indeed we agree that comparing our simple (lumped rainfall-runoff) calibrated simulations to their studies would be benefitial, but we believe it is beyond the scope of the current work. In particular, it is part of our research hypothesis that simple lumped simulation models can be used that do not necessitate an elaborate model setup (in other words that calibrating models in the gauge data period alleviates the inherent model deficiencies). We would further like to point out that some of the models used in the mentioned studies serve other objectives – land surface models are, first of all, developed to reproduce evapotranspiration – which are not required here.

Pedinotti et al (2012) used ISBA-TRIP, a full-fledged continental-scale land surface scheme which was augmented with river routing, floodplain and deep aquifer modules. They evaluate the model with GRACE data, altimetric water heights, in-situ discharge, and satellite-derived flood extent. In Casé et al (we assume the reviewer meant Casse et al, Model-based study of the role of rainfall and land use–land cover in the changes in the occurrence and intensity of Niger red floods in Niamey between 1953 and 2012, in HESS 2016) the same model, ISBA-TRIP is used with various forcing data sets

to study the role of rainfall and land cover on discharge. In both studies, the model appears not to be calibrated to gauge discharge.

In the same spirit, Fleischman et al. (2018) evaluated a two-way coupled hydrological-hydrodynamic model (MGB) in different model structure variants and forced by a single multi-satellite precipitation product. They calibrated their model using observed gauge discharge and validated against altimetric water levels and MODIS flood extent. For those gauges that are used in our study as well (Koulikoro, Ansongo, Dire), they find Nash-Sutcliffe coefficient similar to ours (within 0.1 better or worse). We note they used TMPA precipitation which is close to PERSIANN, the data that we used, but not identical.

Along the same line, we could mention our own recent work in Pomeon et al. (2018). In this work, the SWAT model was set up for a region close to the study region here, calibrated with gauge discharge, and evaluated against GRACE and MODIS evapo-transpiration.

In contrast, Tourian et al (we assume the reviewer meant Tourian M.J., Schwatke C., Sneeuw N.: River discharge estimation at daily resolution from satellite altimetry over an entire river basin. Journal of Hydrology, Vol. 546, 230-247, 10.1016/j.jhydrol.2017.01.009, 2017) develop a purely empirical linear cyclostationary dynamic model for discharge, which is learned by ingesting daily gauge time series and which assimilates altimetric discharge (i.e. levels turned to discharge in a preprocessing step via a modified RC approach). This represents an intelligent interpolation approach for discharge rather than a physical modelling study.

As mentioned above, it could make an interesting comparative study to use modelled discharge from any of these studies for generating rating curves within a common time window. However, while we could probably obtain simulated discharge from the authors of these studies, these models are all less straightforward to set up and to transfer to other regions compared to GR4J and HBVlight, and this would thus not add to the

objective of the present paper.

As a result of this reasoning, we suggest to modify the sentence on line 32 "Others have proposed to assimilate altimetric levels into elaborate hydrodynamic modelling (Munier et al., 2015); however such models are not always available" to "Others have proposed to simulate discharge using fully-fledged calibrated/validated land surface modelling (Pedinotti et al., 2012, Casse et al., 2016, Fleischmann et al., 2018, or Pomeon et al., 2018), assimilate altimetric levels into elaborate hydrodynamic modelling (Munier et al., 2015), or interpolate discharge based on empirical dynamic models trained on gauge discharge (Tourian et al., 2017); however such models are not always available and less straightforward to transfer to new regions."

**RC:** Page 2, line 4 : that "radar retracking is key for obtaining meaningful time series..." has been shown in many publications over more than a decade...

**AR:** This is correct. We modify "we will show" to "we will confirm"

**RC:** Methods and Data Page 5 line 5 : Capability in retrieving water levels has considerably evolved from TP, Jason1 to Jason 2. Hence, they cannot be qualified by a single "70-80cm RMSE". Note that much better results than those by Tourian et al. (2016) can be found in the literature for Jason 2 (see for instance Seyler et al., 2012)

And

**RC:** Line 8: Tourian et al. (2016) estimated their baises over the ocean, which is of limited meaning for rivers because the echoes are too different. other estimates are available in the literature

**AR:** The "70-80cm RMSE" is the Tourian et al estimate for Jason-2, i.e. not a single value for TP, J1 and J2. But we are aware that better results have been reported, e.g. in Seyler et al. (2013) though the authors of this study appear reluctant ("0,35 m . . .could be then the error estimated on the water stage derived from Jason-2 ranges, when no other validation is available."). Papa et al. (2012) suggest 0,28 and 0,19 m for Ganges

and Brahmaputra – clearly this depends on river width. We modify the sentence to "Relative altimeter errors (...) are thought to be at the level of 20-80 cm Root Mean Square Error (RMSE) e.g. for Jason-2 dependent on river width (Papa et al., 2012, Seyler et al., 2013, Tourian et al., 2016)". For Envisat and Saral/Altika: "Relative Errors are believed to be at the 15-70 cm range (Sridevi et al., 2016, Tourian et al., 2016, Bogning et al., 2018)".

**RC:** Line 9-10 : "biases [...] reference systems effects". Could the authors explain which "reference system " may turn into significant biais effect ??

**AR:** The reviewer is right here. What we meant here is that reference system effects (e.g. realization of the vertical datum, connection of gauge to datum, geoid model error) add to observed systematic descrepancies and to the estimated bias, but this is of course not the altimeter's fault. We suggest to remove "reference system effects" for clarity.

**RC:** Page 6 lines 4-5 : Roscher et al. (2017) retracked altimetry data on the coastal domain. I may have missed the publication but I am not aware of any publication showing that their STAR retracking performs better over rivers (both in height accuracy and number of data) than the other-usual- algorithms, such as ICE1, ICE3, ... . If so, authors should add a reference.

**AR:** The reviewer is right; the present study is the first publication where the Roscher et al STAR method is used over rivers. Please note that we do not claim STAR to be the only method which provides decent heights over rivers. Other approaches, such as the Multi-Subwaveform Retracker (MSR, Boergens et al., 2016) often lead to very similar results compared to our STAR approach. From Fig. 1 it becomes obvious that simple retrackers such as ICE1 in this case can lead to unrealistic peaks over rivers, especially during low water level conditions. However, please note that we do not want to provide an in depth discussion of retracking methods here since this is not the main topic of this paper. We modify the sentence to "which lead to much more ... in coastal

applications. Here, we make use of the point-cloud by-product of STAR in order to derive improved river heights. ".

**RC:** Line 8 : Berry presented the hooking effect extensively in meetings in the early 2000's but as far as I know, the first publication presenting the use of these parabolas to gain accuracy in retrieving water levels is Frappart et al. (RSE 2006).

**AR:** True. We add the reference to Frappart et al. (2006).

**RC:** Line 8: "which masks water levels". Actually, since the onboard tracking is locked on the river echo, the bank topography is masked by the river, not the opposite.

**AR:** Yes, this is actually explained with the previous sentence. We shorten this sentence to "leads to the so-called 'hooking effect' (. . .), a spurious parabolic profile . . ."

**RC:** Page 7 line 8 : "... even over narrow river crossing ...". It may be a question of my own limitations in english but I think that "even" should be replaced by "in particular" sine the hooking effect is particulary visible -and useful- over the narrow reaches.

**AR:** This depends on the point of view. The hooking effect is particularly pronounced over narrower crossings and it is thus easier to detect/remove it. In this sense the reviewer is correct (of course if there would be no topography i.e. the river much wider, there would be less hooking effect but nevertheless better altimetric accuracy; this was what we meant to say). To remove this ambiguity, we decide to remove the word "even" from the sentence.

**RC:** Line 20 : Authors mention that the S3A data are distributed with 2 retracking algorithms, the OCOG and the Samosa ones. I suggest that both algorithms will be tested in their study.

**AR:** Actually both have been included already, please see results in Fig. 7 and the explanation on page 7 lines 12-15. GPOD uses Samosa and Copernicus used OCOG.

**RC:** Line 26 : Authors should mantion that the unit of a is $m^{\frac{4}{3}}s^{-1}$ only if $b = \frac{5}{3}$ ( if b is

different from $\frac{5}{3}$, the units would not equilibrate on both sides of Eq 1

**AR:** This is correct. We suggest to modify the sentence from "In Eq. (1), a (in $m^{\frac{4}{3}}s^{-1}$) and the dimensionless parameter b describe the hydraulic behaviour. These parameters can be computed from Manning's equation under idealized conditions (Paris et al., 2016)." To "In Eq. (1), a and b describe the hydraulic behaviour. These parameters can be computed from Manning's equation under idealized conditions (Paris et al., 2016), then usually $b = \frac{5}{3}$ (dimensionless) and a would be in units of $m^{\frac{4}{3}}s^{-1}$."

**RC:** Line 30 : "river width is difficult to observe from space". This sentence should be reworded since river width is definitely not difficult to observe from space and it is commonly derived from existing satellite imagery products

**AR:** River width needs to be known or observed at the time of the altimeter overflight, and for optical sensors this is may be challenging in tropical regions due to cloud coverage. We agree with the new Sentinel products this problem may be largely alleviated, but around 2000 the situation was clearly different from now. We suggest to modify the sentence to "river width has been difficult to observed in the past".

**RC:** Page 8, line 1 : "assuming gauge and virtual gauge data [...] area vailable...". Replace "gauge" by "discharge" ??

**AR:** Thanks for pointing this out. We suggest to modify "Assuming gauge and virtual gauge data from altimetry are available" to "Assuming observed discharge and virtual gauge level data from altimetry are available".

**RC:** Page 9 Eq 3 : I suggest that authors mention that Guetirana  Peters-Lidar (HESS 2013) showed that this methology based on minimum of squared residuals of Log regressions can converge towards physically meaningless values (which may explain the final results in terms of rating curves found in the present study

**AR:** Getirana and Peters-Lidard (HESS 2013) indeed show that minimizing the residual of Eq. (3) (their Eq. (2) is identical to our Eq. (3)) will not necessarily lead to a converging solution for the parameter Z0. However, they also state that non-convergence does not imply bad fitting (i.e.i inability to simulate discharge). It simply mirrors the fact that even Z0 values which are false can produce a better fit than the true one. This leads to a high uncertainty for the final result of Z0 and thus indeed explains our final results in terms of rating curves. We suggest to add an explanation in the discussion rather than in the methodology: "Getirana and Peters-Lidard (2013) point out that the used procedure of rating curve fitting may not nessecarily converge for Z0, which may lead to inaccurate values for this parameter." (section 3.3, after "Noticeable is that the rating curve is almost parallel...").

**RC:** Line 9 : I suggest that the authors present an example of "breakpoint" in a figure.

**AR:** We add a reference here to Fig. 8 (GR4J, Envisat with BP).

Page 17 : all these results, although interesting, lack analysis with respect to literature, in particular with respect to the recent study by Tourian et al.(HESS 2017) and by Fleischman et al. (JoH, 2018).

**AR:** As discussed earlier, these studies had somewhat different objectives and so it is not easy to compare. We suggest to focus here on (1) skill of simulated discharge w.r.t. gauge discharge, (2) altimetric discharge w.r.t. simulated discharge. We add a paragraph that compares our results to the studies of Tourian et al. (2017) and Fleischmann et al. (2018) as far as this is possible.

**RC:** Page 18: Figure 8 is quite difficult to read and I suggest that authors redo it. Actually, it would be more clear to plot the discharge versus the true altimetric height instead of versus the water depth (given that, in addition, no information is provided about the Zo parameters). Plotted this way one would have only 2 discharge values for a given altimetric height. Also, information about the Zo parameters could be presented simply by the intersection of the rating curves with the height axes.

**AR:** We changed the figure in the proposed way (Fig. 2) and adapted the discussion

to it.

**RC:** Page 21, line 21 : Sentence "we find that simulated discharge may aid..." is really over-selling the findings of the present study since many studies already showed such a result. The sentence has to be reworded.

**AR:** We suggest to replace "We find that simulated discharge may aid in creating empirical altimetry-discharge rating curves..." by "We find that discharge simulated by simple lumped rainfall-runoff models may aid in creating empirical altimetry-discharge rating curves..."

**RC:** Line 28: ".... most likely...". In absence of any evidence presented in the study, I think that" presumably" would be better adapted.

**AR:** We agree and we have modified the sentence to "presumably when the river inundates its banks."

**References**

- Bogning, S., Frappart, F., Blarel, F., Niño, F., Mahé, G., Bricquet, J.-P., Seyler, F., Onguéné, R., Etamé, J., Paiz, M.-C., Braun, J.-J., 2018. Monitoring Water Levels and Discharges Using Radar Altimetry in an Ungauged River Basin: The Case of the Ogooué. Remote Sensing 10, 350. https://doi.org/10.3390/rs10020350

- Casse, C., Gosset, M., Vischel, T., Quantin, G., Tanimoun, B.A., 2016. Model-based study of the role of rainfall and land use–land cover in the changes in the occurrence and intensity of Niger red floods in Niamey between 1953 and 2012. Hydrology and Earth System Sciences 20, 2841–2859. https://doi.org/10.5194/hess-20-2841-2016

- Fleischmann, A., Siqueira, V., Paris, A., Collischonn, W., Paiva, R., Pontes, P., Crétaux, J.-F., Bergé-Nguyen, M., Biancamaria, S., Gosset, M., Calmant, S., Tanimoun, B., 2018. Modelling hydrologic and hydrodynamic processes

in basins with large semi-arid wetlands. Journal of Hydrology 561, 943–959. https://doi.org/10.1016/j.jhydrol.2018.04.041

• Frappart, F., Calmant, S., Cauhopé, M., Seyler, F., Cazenave, A., 2006. Preliminary results of ENVISAT RA-2-derived water levels validation over the Amazon basin. Remote Sensing of Environment 100, 252–264. https://doi.org/10.1016/j.rse.2005.10.027

• Papa, F., Bala, S.K., Pandey, R.K., Durand, F., Gopalakrishna, V.V., Rahman, A., Rossow, W.B., 2012. Ganga-Brahmaputra river discharge from Jason-2 radar altimetry: An update to the long-term satellite-derived estimates of continental freshwater forcing flux into the Bay of Bengal. Journal of Geophysical Research: Oceans 117. https://doi.org/10.1029/2012JC008158

• Pedinotti, V., Boone, A., Decharme, B., Crétaux, J.F., Mognard, N., Panthou, G., Papa, F., Tanimoun, B.A., 2012. Evaluation of the ISBA-TRIP continental hydrologic system over the Niger basin using in situ and satellite derived datasets. Hydrology and Earth System Sciences 16, 1745–1773. https://doi.org/10.5194/hess-16-1745-2012

• Poméon, T., Diekkrüger, B., Springer, A., Kusche, J., Eicker, A., 2018. Multi-Objective Validation of SWAT for Sparsely-Gauged West African River Basins—A Remote Sensing Approach. Water 10, 451. https://doi.org/10.3390/w10040451

• Seyler, F., Calmant, S., Silva, J.S. da, Moreira, D.M., Mercier, F., Shum, C.K., 2013. From TOPEX/Poseidon to Jason-2/OSTM in the Amazon basin. Advances in Space Research, Satellite Altimetry Calibration and Deformation Monitoring using GNSS 51, 1542–1550. https://doi.org/10.1016/j.asr.2012.11.002

• Sridevi, T., Sharma, R., Mehra, P., Prasad, K.V.S.R., 2016. Estimating discharge from the Godavari River using ENVISAT, Jason-2, and

SARAL/AltiKa radar altimeters. Remote Sensing Letters 7, 348–357. https://doi.org/10.1080/2150704X.2015.1130876

- Tourian, M.J., Schwatke, C., Sneeuw, N., 2017. River discharge estimation at daily resolution from satellite altimetry over an entire river basin. Journal of Hydrology 546, 230–247. https://doi.org/10.1016/j.jhydrol.2017.01.009

[Figure]

[Figure]

**Fig. 1.** STAR vs. ICE1 retracker

[Figure]

**Fig. 2.** Rating curves with altimetric heights on x-axis

---

## Author Comment (AC3) · 25 Apr 2019

We have two additional, noteworthy points:

1) The reviewer wrote: "Line 15: LRM altimetry has been used to produce hundreds of series over narrow reaches, up to a few tens of meters wide (see for example those in the Amazon basin distributed by hydroweb). I may have missed the publications (and none is given to support the statement) but I did not read that either Cryosat-2 or S3A did much better, up to now. SAR just enables sampling more small reaches than LRM does."

Although publications on CryoSat-2 and Sentinal-3 are still not numerous, Schneider et al. (2018) show that Cryosat-2 outperforms previous altimetric missions for rivers of

average width of about 300 meters, as the Po river. Our group found similar conclusions for the Rhine and Elbe river.

2) The reviewer wrote: "Authors mention that the S3A data are distributed with 2 retracking algorithms the OCOG and the Samosa ones. I suggest that both algorithms will be tested in their study."

We mistakenly understood that the reviewer meant the S3A data from Copernicus Open Access Hub and GPOD. The S3A "distribution", i.e. the Copernicus data, are indeed provided with two different retracking algorithms: OCOG and Samosa-2 – in GPOD, the Samosa+ retracker is used. Up to now, we have not shown the Samosa-2 results because the OCOG gave better results in our case. Following the suggestion of the reviewer, we will comment also on the second retracker.

**Literature:**

Schneider, R., Tarpanelli, A., Nielsen, K., Madsen, H., Bauer-Gottwein, P., 2018. Evaluation of multi-mode CryoSat-2 altimetry data over the Po River against in situ data and a hydrodynamic model. Advances in Water Resources 112, 17–26. https://doi.org/10.1016/j.advwatres.2017.11.027